

# The effectiveness of cooling conditions on temperature of canine EDTA whole blood samples

Karen M. Tobias[1], Leslie Serrano[2], Xiaocun Sun[3] and Bente Flatland[4]

[1] Department of Small Animal Clinical Sciences, College of Veterinary Medicine, University of Tennessee Institute of Agriculture, Knoxville, TN, USA
[2] College of Veterinary Medicine, University of Tennessee Institute of Agriculture, Knoxville, TN, USA
[3] Office of Information Technology, University of Tennessee–Knoxville, Knoxville, TN, USA
[4] Department of Biomedical and Diagnostic Sciences, University of Tennessee Institute of Agriculture, Knoxville, TN, USA

Corresponding author
Karen M. Tobias, ktobias@utk.edu

## ABSTRACT

**Background:** Preanalytic factors such as time and temperature can have significant effects on laboratory test results. For example, ammonium concentration will increase 31% in blood samples stored at room temperature for 30 min before centrifugation. To reduce preanalytic error, blood samples may be placed in precooled tubes and chilled on ice or in ice water baths; however, the effectiveness of these modalities in cooling blood samples has not been formally evaluated. The purpose of this study was to evaluate the effectiveness of various cooling modalities on reducing temperature of EDTA whole blood samples.

**Methods:** Pooled samples of canine EDTA whole blood were divided into two aliquots. Saline was added to one aliquot to produce a packed cell volume (PCV) of 40% and to the second aliquot to produce a PCV of 20% (simulated anemia). Thirty samples from each aliquot were warmed to 37.7 °C and cooled in 2 ml allotments under one of three conditions: in ice, in ice after transfer to a precooled tube, or in an ice water bath. Temperature of each sample was recorded at one minute intervals for 15 min.

**Results:** Within treatment conditions, sample PCV had no significant effect on cooling. Cooling in ice water was significantly faster than cooling in ice only or transferring the sample to a precooled tube and cooling it on ice. Mean temperature of samples cooled in ice water was significantly lower at 15 min than mean temperatures of those cooled in ice, whether or not the tube was precooled. By 4 min, samples cooled in an ice water bath had reached mean temperatures less than 4 °C (refrigeration temperature), while samples cooled in other conditions remained above 4.0 °C for at least 11 min. For samples with a PCV of 40%, precooling the tube had no significant effect on rate of cooling on ice. For samples with a PCV of 20%, transfer to a precooled tube resulted in a significantly faster rate of cooling than direct placement of the warmed tube onto ice.

**Discussion:** Canine EDTA whole blood samples cool most rapidly and to a greater degree when placed in an ice-water bath rather than in ice. Samples stored on ice water can rapidly drop below normal refrigeration temperatures; this should be taken into consideration when using this cooling modality.

## INTRODUCTION

Laboratory error can occur in the pre-, intra-, and postanalytic phases of sample analysis, with 31.6–75% of errors occurring in the preanalytic phase (*Bonini et al., 2002*). Examples of preanalytic errors include lack of patient fasting, iatrogenic hemolysis, mislabeling of samples, and inappropriate sample storage. Measurement of whole blood and plasma ammonium concentration is useful for diagnosis of portovascular anomalies and hepatic encephalopathy and for monitoring response to treatment in people and dogs (*Barsotti, 2001*; *Lidbury et al., 2015*; *Natesan, Mani & Arumugam, 2016*). Ammonium is very labile, and time and temperature have significant preanalytic effects on laboratory results in people (*Heins, Heil & Withold, 1995*; *Howanitz et al., 1984*). For example, storage of whole blood at 4 and 22 °C for 1 h results in increases in plasma ammonium of 31.7 and 39.9%, respectively (*Howanitz et al., 1984*). Heparinized whole blood samples stored in ice water for 30 min before centrifugation have significantly less increases in plasma ammonium than those stored at room temperature for 30 min before centrifugation (13 vs. 31%, respectively) (*Nikolac, Omazic & Simundic, 2014*). Recommendations for handling samples for ammonium measurement include collecting the blood in a stoppered ammonia-free vacuum tube, placing the tube immediately in an ice or ice water bath, and separating the plasma from the sample within 15 min of collection to prevent spontaneous ammonium generation (*Barsotti, 2001*; *Nikolac, Omazic & Simundic, 2014*; *Association for Clinical Biochemistry, 2012*; *Dukic & Simundic, 2015*; *Maranda et al., 2007*). Some authors also recommend using a precooled vacuum sample tube to speed the specimen cooling process (*Barsotti, 2001*; *Maranda et al., 2007*). Effectiveness of precooling tubes and of use of ice versus ice water baths for cooling blood samples has not been formally evaluated and reported in the medical literature. The purpose of this study was to compare different sample cooling protocols to determine which would be most effective for rapidly reducing blood sample temperature before centrifugation and analysis. This work is part of an effort to optimize sample handling protocols for ammonium analysis in our hospital.

## MATERIALS AND METHODS

Leftover refrigerated canine EDTA whole blood samples were obtained from the University of Tennessee, College of Veterinary Medicine Clinical Pathology Laboratory and used for testing. All samples were seven days old at the time of entry into the study and had been stored at 4 °C. Samples were pooled, and a packed cell volume (PCV) of the resulting pool was measured. The pool was then divided into two aliquots. For aliquot 1 (normal PCV), physiologic saline was added to decrease the PCV to 40%. For aliquot 2 (simulated anemia), physiologic saline was added to decrease the PCV to 20%. Each aliquot was divided into 2 ml samples and 2.5 ml samples that were placed in a 13 × 75 mm, stoppered, EDTA-free sample tubes (Vacutainer®; Becton Dickinson

Company) and kept refrigerated at 4 °C until testing. In all test conditions, the cooling rate of 2 mls of blood was to be evaluated; however, placement of a warmed blood sample into a precooled tube required transfer of the blood. For those samples, an extra 0.5 ml of blood was included in the storage tubes to ensure a full 2 mls of blood was transferred at the time of testing. Two mls of blood was chosen because it is the standard volume collected in a 13 × 75 mm tube containing 3.6 mg of potassium EDTA; additionally, that volume of blood forms a column less than 30 mm in height within the tube, ensuring that the surface level of the blood will rest several cm below the surface level of the ice or ice/water.

The testing apparatus consisted of a lidded 148 ml plastic specimen cup (Fisherbrand™ multipurpose specimen storage containers; Fisher Scientific), a temperature probe with an accuracy of ±0.3 °C (Fisher Scientific™ Traceable™ Hi-Accuracy Thermometer), and a support stand with clamps. A whole the diameter of the sample tube was drilled in the lid of the specimen cup to allow the tube to be inserted through the lid and to be held upright and centered in the specimen cup with the bottom of the tube touching the bottom of the cup and the stopper resting above the surface of the lid. A hole was drilled in a tube stopper, which was secured onto the temperature probe and used for all measurements, so that the temperature probe would maintain the same location and depth in each tube and so that each tube remained stoppered for the entirety of the test.

Immediately before testing, stoppered blood tubes were placed on a rack in a 38 °C water bath and warmed to 37.7 °C. Samples from each aliquot (normal PCV or anemic PCV) were tested under one of three cooling conditions: 1, Ice bath; 2, Precooled (P/C) tube in ice bath; and 3, Ice water bath. For Condition 1 (ice bath), the specimen cup was filled with 70 gm of crushed ice and capped with the prefabricated lid. A 13 × 75 mm stoppered sample tube containing 2 mls of warmed (37.7 °C) blood was inserted through the hole in the lid, and its stopper was replaced with the stopper-temperature probe unit. For Condition 2 (P/C tube in ice bath), an empty 13 × 75 mm stoppered sample tube was placed on ice. After 5 min, 2 mls of blood were withdrawn from a storage tube containing 2.5 mls of warmed (37.7 °C) blood and transferred into the P/C sample tube, which was then positioned in a cup containing 70 g of crushed ice and capped with the stopper-temperature probe unit, as for Condition 1. For Condition 3 (Ice water bath), the specimen cup was filled with 70 g of crushed ice and 45 g of water and covered with the prefabricated lid. Immediately thereafter, a 13 × 75 mm sample tube containing 2 mls of warmed blood was capped and positioned in the specimen cup, as described for Conditions 1 and 2. In each condition, the surface of the blood sample rested several cm below the surface of the cooling agent (ice or ice water). Temperature of each sample was recorded at one-minute intervals for 15 min.

Six treatments (ice bath/PCV 20%; ice bath/PCV 40%; P/C in ice/PCV 20%; P/C in ice/PCV 40%; ice water/PCV 20%; ice water/PCV 40%) were evaluated over 16 time points (0–15 min), with 30 replicates per treatment. Temperature data were evaluated using a one-way ANOVA with repeated measures, with treatment as the between subject factor and time as the within subject factor. Time was analyzed as a categorical variable and a numeric variable, respectively, to estimate the effect of treatment for each time

 

**Table 1 Summary of treatment comparisons (treatment 1 vs. treatment 2) showing effectiveness of blood sample cooling conditions on the mean temperature drop over 15 min.**

|  | Ice 20% PCV | Ice 40% PCV | P/C 20% PCV | P/C 40% PCV | Water/Ice 20% PCV | Water/Ice 40% PCV |
|---|---|---|---|---|---|---|
| Ice 20% PCV | – | p = 0.6405 | p < 0.0001 | – | p < 0.0001 | – |
| Ice 40% PCV | p = 0.6405 | – | – | p = 0.7974 | – | p < 0.0001 |
| P/C 20% PCV | p < 0.0001 | – | – | p = 0.1697 | p < 0.0001 | – |
| P/C 40% PCV | – | p = 0.7974 | p = 0.1697 | – | – | p < 0.0001 |
| Water/Ice 20% PCV | p < 0.0001 | – | p < 0.0001 | – | – | p = 0.9983 |
| Water/Ice 40% PCV | – | p < 0.0001 | – | p < 0.0001 | p = 0.9983 | – |

Notes:
Ice, ice bath; P/C, precooled tube in ice bath; water/ice, ice water bath; PCV, packed cell volume.
p value = Tukey's adjusted probability value, with significance set at p < 0.05.

interval and the overall trend of time effect. The rate of cooling (speed) data was analyzed using one-way ANOVA. Rate of cooling was computed using the range of the temperature change divided by the total time interval (15 min). Ranked transformation was applied when data violated ANOVA assumptions such as non-normality and unequal variance. Post hoc multiple comparison was conducted with Tukey's adjust. To analyze the trend of time, time was included in the model as a numeric factor whereas treatment was included as a categorical variable. Significance was set at p < 0.05, and all analysis was processed in SAS9.4 TS1M1 for Windows x64 (Cary, NC).

## RESULTS

Results are summarized in Tables 1 and 2. Time exhibited a significant decreasing effect (p < 0.0001) on temperature when controlling for the treatment effect. However, time and treatment interaction effect was also significant (p < 0.0001), indicating the effect of time was dependent on treatment.

Within treatment conditions (ice, ice water, and P/C tube on ice), sample PCV had no significant effect on cooling. Cooling in ice water was significantly faster than cooling on ice only or P/C tube on ice (Fig. 1). Regardless of PCV, mean temperatures of samples cooled in ice water were significantly lower at 15 min than mean temperatures of samples cooled on ice only or in P/C tube on ice (p < 0.0001). By 4 min, samples cooled in ice water had reached mean temperatures less than 4 °C (that of refrigeration), regardless of PCV.

For samples with a 20% PCV, mean temperature at 4 min of cooling in ice water was significantly lower than mean temperature at 14 min for samples cooled on ice only (p = 0.0148) or at 11 min for samples in P/C tube on ice (p < 0.0001). Of samples with a 40% PCV, mean temperature at 4 min of samples cooled in ice water was significantly cooler than mean temperature at 13 min for samples cooled on ice only (p = 0.0335) or at 12 min for samples in P/C tube on ice (p < 0.0001).

For samples with a PCV of 40%, there was no difference in mean temperatures or rate of cooling of samples placed on ice and those placed in P/C tubes on ice. For samples with a 20% PCV, however, the rate of cooling was significantly faster for samples in a P/C tube on ice versus those placed on ice. From 4 to 13 min, mean temperatures of samples

**Table 2 Mean temperature (°C) per minute of normal and simulated anemic blood samples for three cooling conditions.**

| Time (MIN) | Ice 20% PCV | Ice 40% PCV | Precooled 20% PCV | Precooled 40% PCV | Water/Ice 20% PCV | Water/Ice 40% PCV |
|---|---|---|---|---|---|---|
| 0 | 37.7 | 37.7 | 37.7 | 37.7 | 37.7 | 37.7 |
| 1 | 27.74 | 27.6 | 22.57 | 23.91 | 17.53 | 19.07 |
| 2 | 21.94 | 21.88 | 17.9 | 19.74 | 9.26 | 10.09 |
| 3 | 18.35 | 17.93 | 14.68 | 16.41 | 5.26 | 5.53 |
| 4 | 15.5 | 15.07 | 12.23 | 13.76 | 3.07 | 3.09 |
| 5 | 13.22 | 12.75 | 10.36 | 11.68 | 1.84 | 1.78 |
| 6 | 11.4 | 10.85 | 8.86 | 9.99 | 1.12 | 1.06 |
| 7 | 9.88 | 9.27 | 7.65 | 8.64 | 0.71 | 0.66 |
| 8 | 8.62 | 7.99 | 6.63 | 7.51 | 0.47 | 0.44 |
| 9 | 7.54 | 6.94 | 5.78 | 6.56 | 0.33 | 0.31 |
| 10 | 6.65 | 6.05 | 5.06 | 5.74 | 0.24 | 0.24 |
| 11 | 5.88 | 5.3 | 4.46 | 5.05 | 0.20 | 0.2 |
| 12 | 5.22 | 4.67 | 3.98 | 4.45 | 0.17 | 0.18 |
| 13 | 4.65 | 4.12 | 3.5 | 3.94 | 0.16 | 0.16 |
| 14 | 4.14 | 3.66 | 3.12 | 3.51 | 0.15 | 0.16 |
| 15 | 3.69 | 3.27 | 2.8 | 3.14 | 0.14 | 0.15 |

Note:
  PCV, Packed cell volume; MIN, minutes.

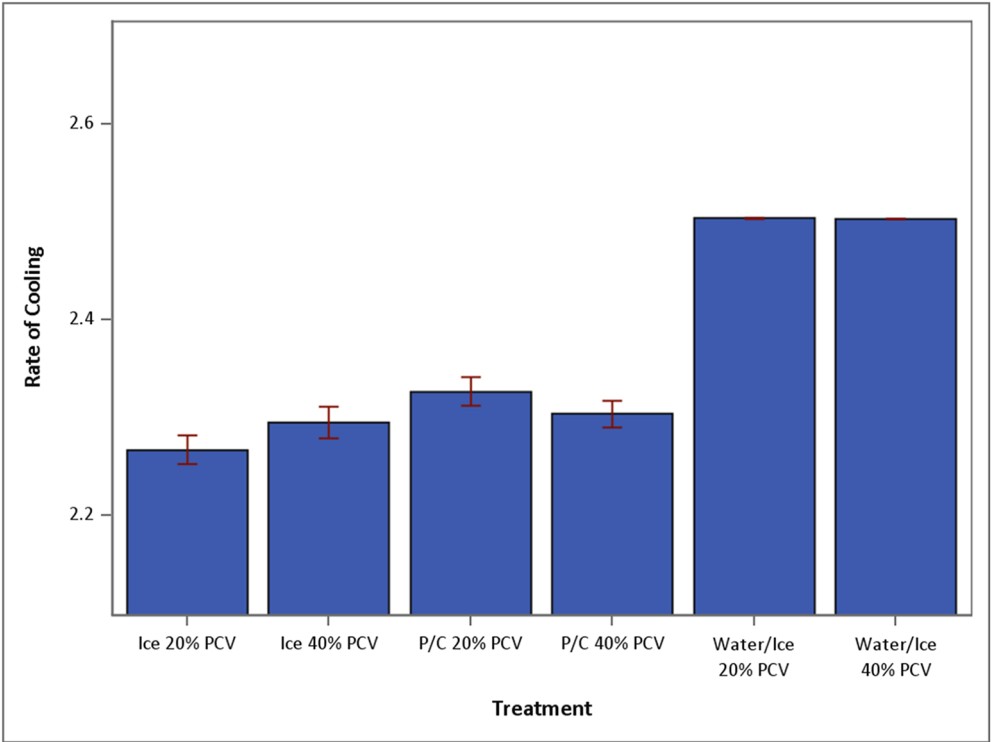

**Figure 1 Mean speed of temperature drop over 15 min (+SEM) under conditions of ice 20%, ice 40%, P/C 20%, P/C 40%, water/ice 20% and water/ice 40% (N = 30 for each treatment group).** Ice, ice bath; P/C, precooled tube in ice bath; Water/Ice, ice water bath.

with a 20% PCV that were placed in a P/C tube on ice were significantly lower than mean temperatures of samples placed on ice without precooling the tube.

## DISCUSSION

In this study, we determined blood sample cooling occurred more rapidly and to a greater extent when samples were placed in an ice water bath versus in a cup of crushed ice. With a closed system that lacks motion, as described in this study, cooling of blood samples relies primarily on conduction. Conduction is the flow of heat that reduces the temperature difference between two materials (*Radi & Rasmussen, 2013*). Transfer of heat via conduction relies on exchange of kinetic energy between microscopic particles within substances: in this case, between the blood and the tube containing it, and between the tube and its external environment (ice or ice water). Variables important to heat transfer include the material's innate ability to conduct heat (thermal conductivity), the thickness of the materials involved, the surface area available for heat transfer, and the temperature difference across these surfaces (*Radi & Rasmussen, 2013*; *Vlachopoulos & Strutt, 2002*). In our study, sample volume and tube size and thickness were standardized; therefore, differences in rates of cooling were primarily dependent on the thermal conductivity of the materials used for cooling, the amount of contact each of these substances has with the tube, and the temperature of the cooling condition. Rate of cooling may change if the tube contains more or less blood or if the surface level of the blood is not totally submerged below the surface level of the ice or ice water.

In our study, cooling was performed using closed systems containing crushed ice or ice water. Ice water consists of two major components, ice and water, with thermal conductivity coefficients of 2.1 W/(m K) at 0 °C for ice and 0.6 W/(m K) at 25 °C for water. Crushed ice contains a small amount of liquid water but primarily consists of ice and air, the latter of which has a thermal conductivity coefficient of 0.024 W/(m K) at 0 °C. Because thermal conductivity of water is much greater than air, it serves as a better conductor of heat and thus increases the rate of heat loss from the sample (*Vlachopoulos & Strutt, 2002*).

Surface areas of the cooling materials used in this study cannot be calculated, since pieces of crushed ice varied greatly in size and shape and thus in their ability to contact the surface of the tube. Therefore, a variable amount of air was present within the crushed ice condition. Addition of water to the ice increased the surface area for diffusion of heat via a material with higher conductivity, thereby speeding heat transfer. Temperature of the condition (ice or ice water) was not measured in this experiment. Temperature difference was likely greater between the sample and the ice water, however, since pockets of air within the crushed ice could have maintained higher temperatures than the ice water.

A PCV of 40% was chosen as the normal PCV in this study because it was consistent with the mean PCV (39.6% ± 6.4) of 102 dogs presenting for ammonium measurement at our clinic (K. Tobias, 2016, unpublished data). The samples were diluted with physiologic saline (versus plasma) to maintain consistency, since plasma components can vary among dogs. Between conditions, PCV had no significant effect on rates of cooling.

This is not surprising, since the coefficients of thermal conductivity for EDTA blood and EDTA plasma are approximately the same as that of water (*Ponder, 1962*). However, while no significant difference was noted between cooling on ice and use of a precooled tube plus cooling on ice for samples with a PCV of 40%, use of a precooled tube produced a significantly greater reduction in temperature for samples with a PCV of 20%. Rather than being an effect of PCV per se, a possible cause for this finding is an effect of tube handling. To mimic clinic conditions used in sample collection for ammonium analysis, precooling was performed by placement of a stoppered tube on ice. Leaving the stopper in place during sample collection and transport prevents inadvertent contamination of the tube with environmental factors, such as cigarette smoke or cleaning agents, that can result in erroneously increases in measured ammonium (*Hashim & Cuthbert, 2014*). However, in our system the stoppered tubes were manually removed from the ice bath to have their caps removed, which could have increased temperature of the tube by introduction of warmer environmental air and by heat transfer from the hand to the tube surface. Although it is unlikely, we cannot rule out that tubes containing blood with a 40% PCV were handled more or kept out of the ice bath longer during cap removal and tube filling, resulting in a smaller temperature difference between the sample and the tube and a slower rate of cooling.

One concern with use of an ice water bath for cooling of diagnostic whole blood specimens is whether the rapid rate of cooling and attainment of "near-freezing" temperatures will adversely affect samples. In terms of ammonium concentration, sample storage at 0 to −80 °C will reduce, but not halt, spontaneous ammonia generation (*Howanitz et al., 1984*; *Da Fonseca-Wolheim, 1990*; *Da Fonseca-Wolhelm, 1990*). Both rapid increases in temperature and intracellular formation of ice crystals can increase red cell fragility, which can result in hemolysis and increased plasma ammonium concentrations (*Nikolac, Omazic & Simundic, 2014*; *Maranda et al., 2007*; *El-Khoury, Bunch & Wang, 2012*; *Patterson et al., 2011*). In one study, rate of ammonia increase in whole blood significantly correlated with erythrocyte count and plasma alanine aminotransferase and gamma glutamyltransferase concentrations when blood was stored at 20 °C but not when stored at 0 °C (*Da Fonseca-Wolhelm, 1990*). Although hemolysis was not specifically investigated in that study, those findings argue that RBC-related increases in ammonium concentration may be related to cell metabolism rather than hemolysis. Hemolysis was not evaluated in our study, and further study of whether rapid specimen cooling in an ice water bath induces hemolysis, and to what magnitude, is warranted.

## CONCLUSIONS

Canine EDTA whole blood samples cool most rapidly and to a greater degree when placed in an ice water bath rather than in ice. Use of ice water baths may be important for reducing preanalytical error in the measurement of ammonium and other analytes altered by metabolic processes that continue within blood samples after they are collected; further research is needed in this area. Samples stored on ice water can rapidly drop below normal refrigeration temperatures; this should be taken into consideration when using this cooling modality.

### Funding

Funding was provided by the University of Tennessee, College of Veterinary Medicine Center of Excellence. The funders had no role in study design, data collection and analysis, decision to publish, or preparation of the manuscript.

### Grant Disclosures

The following grant information was disclosed by the authors:
University of Tennessee, College of Veterinary Medicine Center of Excellence.

### Competing Interests

The authors declare that they have no competing interests.

### Author Contributions

- Karen M. Tobias conceived and designed the experiments, contributed reagents/materials/analysis tools, wrote the paper, prepared figures and/or tables, reviewed drafts of the paper.
- Leslie Serrano performed the experiments, contributed reagents/materials/analysis tools, wrote the paper, prepared figures and/or tables.
- Xiaocun Sun analyzed the data, wrote the paper, prepared figures and/or tables, reviewed drafts of the paper.
- Bente Flatland conceived and designed the experiments, wrote the paper, reviewed drafts of the paper.

### Data Deposition

 The raw data has been supplied as Supplemental Dataset Files.

### Supplemental Information

Supplemental information for this article can be found online at http://dx.doi.org/10.7717/peerj.2732#supplemental-information.

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
