# Peer review of "The effectiveness of cooling conditions on temperature of canine EDTA whole blood samples"

_PeerJ, doi:10.7717/peerj.2732_

## Round 0.1 · original submission · Major Revisions

You have a goodly number of suggestions for improving your manuscript from experts in veterinary clinical pathology, and I welcome you to submit a revised manuscript provided you can address each of these comments (including those embedded in the annotated manuscript from Reviewer 4).

Reviewer 1 ·

Basic reporting

Figure 1 is superfluous since the data is presented in Table 2, and so should be removed.
Figure 2 – authors should state how ‘rate of cooling’ was calculated. Also should stipulate what the chart shows – is it mean + SD or SEM for example?

Experimental design

No comments - this was appropriate

Validity of the findings

No comments

Additional comments

There is an extra ‘.’ At time = 2 mins in Table 1. (9..26)

Information in Table 1 might be better presented as a matrix than as a table.

Reviewer 2 ·

Basic reporting

The paper is well written.

Title is misleading and should be changed to: “The effectiveness of cooling conditions”, not “the effects..”.

Line 184: Change ammonia to ammonium (consistency throughout the paper)

Line 263 (table 1): change to 'effectiveness' (instead of effect)

Table 2: Water/ice 20%, in the minute 2 column: typographical error (should be 9.26).

Figure 2: The figure legend does not explain how the rate was calculate and it does not state what does the whiskers represent (SD ? SEM?). Also, I don’t believe it adds to the data that are represented in figure 1.

Experimental design

1. The novelty and significance of this paper are highly questionable. The effect of cooling on blood is important. The effectiveness of cooling is only important if the effect of cooling on blood is significant. The authors attempted to show in the introduction why the effect of cooling is important but without showing difference in this effect between different cooling methods, the significance of this paper is questionable.

As the authors indicate in the discussion, the thermal conductivity of water, ice and air are known so there is no novelty per se in demonstrating differences in the rate of cooling among these 3 treatments. The authors also recognized in the discussion that because the thermal conductivity of plasma and whole blood are known to be virtually the same, the comparison of “anemic” blood to “normal” blood is not interesting. This brings up a key point that the authors do not discuss: Is there any novelty in doing this experiment on canine blood? Or is this just an exercise in thermal physics that could have been done while there was any other liquid in the tubes?

2. Blood tubes were stored in the fridge and then the tubes were all warmed up in a 38C water bath. However, the blood in these tubes was not transferred into room-temperature tubes. Instead, the tubes, as is, were placed on ice or ice water. This doesn’t mimic real-life blood collection in which blood in 38C is collected into tubes that are in room temperature. Here, the tubes themselves had a temp of 38C so the time it would take them to cool in ice/water-ice would be different compared to real life situation in which the blood already cools a little bit just by being placed in a tube that is in room temp. While this still technically allows for a comparison between ice-water and ice (although this comparison does not reflect on real life blood cooling), it does not allow for comparison to the third group in which the blood was transferred to tubes that were pre-chilled. Because blood was moved from one set of tubes into “pre-chilled” tubes, this treatment cannot be compared to tubes that are themselves at 38C.
To fix this, blood should have been warmed to 38C but then all samples transferred to room-temp tubes except for those that are pre-chilled.

3. As the authors note in the discussion, the surface area of the crushed ice as well as the temperature of each “condition” were not measured in this study. This is a limitation that wouldn’t have been detrimental if those unknown were at least standardized. However, because they were not, the results here could be spurious, related more to variation in conditions between repeats rather than related to differences between conditions. Similarly, the authors recognize in the discussion that the lack of standardization of the time it took to handle each tube could have affected the results despite the fact that the differences were probably very small. All these seemingly small effects on standardization are important here because the results indicate very small (albeit statistically significant) differences in the temperature and its rate of change. For the very least, the temperature of the ice should have been recorded at the moment of tube placement.

Validity of the findings

Line 211: “Use of ice water baths is critical for reducing preanalytical error in the measurement of ammonium and other analytes altered by metabolic processes that continue within blood samples after they are collected”. This is an overstatement; the authors did not measure ammonium or any other analyte concentration to prove this.

Stat analysis is inappropriate: A repeated measure ANOVA assumes that the repeats are independent. This is not the case when the “repeats” are consecutive time points. In a repeated measures ANOVA there is no significance to the order of treatments but this is definitely not the case when comparing time points. A model taking into account the relationship between time points should be used. Similarly, Post hoc multiple comparisons using Tukey’s HSD also assumes independence of observations which is not the case when comparing time points.

Additional comments

I believe that the paper would have been stronger if the effect on a specific analyte (i.e ammonium) was measured.

Reviewer 3 ·

Basic reporting

Overall the manuscript addresses one of the most basic but important preanalytic variable in blood sample laboratory test - "temperature". Although the scientific community understands the importance of cooling temperature in protocol preparation, proper evaluation of its effect will be helpful.
There are few minor concerns such as :
One concern is that in line 181, 20% PCV is affected by pre-cooled tubes compared to 40% PCV but a proper explanation was not provided. Environmental factors like smoke and cleaning agents should not be responsible for this since the experiments are conducted in a sterile and clean laboratory condition. Kindly address this properly.
There is a word missing in line 141 supposedly.

Experimental design

Line 95, why was 38C used instead of 37C ?

Validity of the findings

In Figure 1 and 2, although ANOVA and statistical analysis was carried out, "stars" to indicate the significance value was missing. Kindly add that in the main figure. Also mention the number of replicates for each experiment in the "materials and methods" section.

Additional comments

Refer to the concerns indicated above.

Reviewer 4 ·

Basic reporting

The manuscript is concise and well written.
As their study relates to canine whole blood samples, it is recommended that the authors specify what species a reference refers to eg. - lines 56 and 59 in the introduction should state " in people" as refs 6 and 7 are human studies. (see uploaded pdf)

Table 1 - the heading refers to rate of blood sample cooling. This needs to be defined - mean temperature drop per minute ? mean temperature drop over 15 minutes ?

t value (presumably Tukey's adjust value) and p value needed to be defined in the footer. Should be lower case v for value in t Value in the column heading.

Figure 1 is considered redundant and should be omitted. It is basically a graphical representation of table 2 but is less informative.

Figure 2. The bars need to be explained in the footer- SD ? SEM ? The Y axis needs to be slightly expanded so that the water / ice 20% and water / ice 40% columns are not compressed at the top with compression of the error bars

Experimental design

For the most part, the study is well designed and executed.

However, it is not clear why the investigators diluted the original pooled blood sample to a Hct of 40%, which is the low end of the reference interval for dogs. It may be that this simplified the "simulated anemia" sample (simple 1:1 dilution) but a 20% Hct for simulated anemia is a completely arbitrary number. Regardless, the original sample did not need to be diluted at all in order to be able to create a 20% Hct "simulated anemia" sample. So, why was this done ? The rationale should be included in the materials and methods. Additionally, the investigators used physiologic saline to dilute both samples - why ? This alters the composition compared to "normal" blood. Could the dilution of plasma protein have an effect on cooling ? This should be addressed in the discussion. Plasma could have been used as a diluent.

Also, the investigators added 2 ml of blood to 7 ml, stoppered, EDTA-free sample tubes. Why ? Presumably because this simplified the testing apparatus (easier to insert and immobilize a 7 ml tube). However, it does not recapitulate reality where complete tube filling is recommended. 7 ml tubes have different wall thickness to 2 ml tubes (thickness of materials is a variable that impacts cooling, as per their discussion). Additionally, 2 ml in a 7 ml tube has a different surface area of sample contact versus 2 ml in a 2 ml tube. This, and potential effects, need to be discussed.

Was there a time delay / equilibration time after the 45g of water was added to the 70g of crushed ice ? (lines 105, 106) and before the tube was added to the cup ? If there was, this should be stated. If not, then it should be stated that the tube was immediately added after the addition of the water.

Validity of the findings

The second sentence of the conclusion should be restated and "softened" (see attached). The effect of an ice-water bath versus ice on pre-analytical error of ANY analyte was not tested in this study. Therefore, it is not known if it is "critical".

Additional comments

This study addresses a salient issue with respect to sample handling and potential pre-analytical error. It confirms something that is intuitive and has been known to food scientists for a long time - that ice water or ice slurries cool faster than ice by virtue of increased surface area of contact. Surprisingly, there is little to no scientific confirmation of this in the context of biological samples such as blood. The authors also address the effects of tube pre-cooling in their study. It is a limited but pragmatic study that provides some useful information with respect to sample handling of blood tubes post collection.

Some additional information and clarifications are considered necessary in order to remove ambiguities and clarify details in order to more fully inform readers - please see the basic reporting, experimental design and validity of the findings sections.

Please also see some additional suggested minor edits in the attached pdf.

Annotated reviews are not available for download in order to protect the identity of reviewers who chose to remain anonymous.

---

## Round 0.2 · Minor Revisions

Please address the following remaining issues with your manuscript:

1. Please ensure consistency by having all P-values with a zero before the decimal place (e.g., P<0.05, not P<.05).

2. The use of the words "effects" and "effectiveness" is inconsistent in the manuscript. The Abstract states that the purpose of this study was to value the effectiveness ..." and Table 1 states: " ... effectiveness of blood sample colling conditions on the mean temperature drop." Your title should therefore contain the word "effectiveness" rather than "effects" (i.e., " ... effectiveness of cooling conditions on temperature of EDTA whole canine blood samples."

3. Please replace the current version of Table 1 with the easier-to-read matrix version in your previous rebuttal to the reviewers. It is not required that you include the t test statistic values, but if you do you should also present the degrees of freedom associated with each t-test.

---

## Round 0.3 · accepted · Accept

Thank you for making the final few requested changes to your manuscript.